# Effect of a Mycotoxin Binder (MMDA) on the Growth Performance, Blood and Carcass Characteristics of Broilers Fed Ochratoxin A and T-2 Mycotoxin Contaminated Diets

**DOI:** 10.3390/ani11113205

**Published:** 2021-11-10

**Authors:** Insaf Riahi, Antonio J. Ramos, Jog Raj, Zdenka Jakovčević, Hunor Farkaš, Marko Vasiljević, Anna Maria Pérez-Vendrell

**Affiliations:** 1Applied Mycology Unit, Food Technology Department, University of Lleida, UTPV-XaRTA, AGROTECNIO-CERCA Center, Av. Rovira Roure 191, 25198 Lleida, Spain; insaf.riahi1@gmail.com (I.R.); antonio.ramos@udl.cat (A.J.R.); 2Patent Co, DOO., Vlade Cetkovica IA, 24211 Subotica, Serbia; jog.raj@patent-co.com (J.R.); zdenka.jakovcevic@patent-co.com (Z.J.); hunor.farkas@patent-co.com (H.F.); marko.vasiljevic@patent-co.com (M.V.); 3Animal Nutrition Department, Institute of Agrifood Research and Technology (IRTA Mas Bové), 43120 Constanti, Spain

**Keywords:** ochratoxin A, T-2 mycotoxin, broiler chickens, mycotoxin binder

## Abstract

**Simple Summary:**

The contamination of feed with mycotoxins is a global concern, resulting in adverse effects on productivity and animal health and, therefore, a great economic loss. Ochratoxin A and T-2 mycotoxins are among the mycotoxins that contaminate animal feed. These mycotoxins could adversely affect the health of broilers, and the most effective method to mitigate the toxic effects of mycotoxins is the use of detoxifying agents. In the present experiment, broiler chickens were allotted into five groups. Group 1 received a non-contaminated diet; group 2 received a non-contaminated diet + 3 g/kg of a mycotoxin binder (MMDA); group 3 received a non-contaminated diet + 0.5 mg/kg OTA + 1 mg/kg T-2 toxin; group 4 received a non-contaminated diet + 0.5 mg/kg OTA + 1 mg/kg T-2 toxin + 1 g/kg MMDA; and group 5 received a non-contaminated diet + 0.5 mg/kg OTA + 1 mg/kg T-2 toxin + 3 g/kg MMDA for 35 days. The results revealed that OTA and T-2 toxin negatively affected the productive parameters and some blood and carcass characteristics of broiler chickens. The addition of the detoxifying agent (MMDA at 1 or 3 g/kg feed) to contaminated diets alleviated the adverse effects observed on productivity and the broilers heath related parameters.

**Abstract:**

The present study was conducted to evaluate the efficacy of the feed additive, a novel multicomponent mycotoxin detoxifying agent (MMDA) containing modified zeolite (clinoptilolite), *Bacillus subtilis*, *B. licheniformis*, *Saccharomyces cerevisiae* cell walls, and silymarin, as detoxifiers of 0.5 mg/kg (0.5 ppm) ochratoxin A (OTA) and 1 mg/kg (1 ppm) T-2 toxin on broiler chickens. A total of 240 1-old broiler chickens (Ross 308) were randomly distributed into five different dietary treatments: (1) control (non-contaminated diet); (2) non contaminated diet + 3 g/kg of MMDA; (3) non-contaminated diet + 0.5 mg/kg OTA + 1 mg/kg T-2 toxin; (4) non-contaminated diet + 0.5 mg/kg OTA + 1 mg/kg T-2 toxin + 1 g/kg MMDA; and (5) non-contaminated diet + 0.5 mg/kg OTA + 1 g/kg T-2 toxin + 3 g/kg MMDA. The results showed that, in the starter period, from 1 to 10 days, the presence of OTA and T-2 mycotoxins reduced the consumption of feed and the growth of the broilers, and no effects of the detoxifying product were observed in the productivity of the chickens, at any of the doses tested, compared to the contaminated control (treatment 3). However, in the growing period, the same negative effect of mycotoxins was registered, but a recovery was observed in the consumption of feed and in the weight of the broilers that consumed 3 g/kg of the MMDA mycotoxin binder, reaching similar values to those of chickens fed uncontaminated control diets. The presence of mycotoxins in feed led to a reduction in the concentration of total proteins and albumin in blood compared to controls, and the presence of the detoxifying product partially reversed this effect. The breast yield of the chickens fed with mycotoxins was lower than that of the animals fed with the control feed and was not affected by the presence of the product tested, at 1 or 3 g/kg. The weight of the different organs (liver, gizzard, kidneys, or spleen), the intestinal pH, the histology of the small intestine, and oral lesions were not affected by the experimental treatments. In summary, the productive parameters and some blood and carcass characteristics of broiler chickens were impaired by the dietary presence of OTA and T-2 toxin. The tested product included at 1 or 3 g/kg feed in contaminated diets improved performance and seems to be effective in partly counteracting the deleterious effects of the tested mycotoxins.

## 1. Introduction

Ochratoxin A (OTA) is a naturally occurring mycotoxin produced by the *Aspergillus* and *Penicillium* species that can be found as a contaminant of poultry feeds. OTA contamination can occur from cool temperate to tropical regions (Northern and Southern America, Northern and Western Europe, Africa and South Asia) [1]. 

Dietary contamination by OTA also poses a big risk for animal health and is a food safety concern due to the transfer of this mycotoxin to humans through the food chain [2,3]. The European Commission provides a maximum guidance level for OTA of 0.1 mg/kg in broiler chickens’ complete feed [4].

Despite poultry being less sensitive to OTA than swine, possibly due to the higher capacity of OTA excretion compared to other species [5], several reports indicate adverse effects of dietary OTA contamination in broiler chicken performances, such as reduced feed intake, body weight gain, and feed efficiency [6,7]. Furthermore, OTA negatively affects the health status of poultry by the alteration of the biochemical, hematological, and histopathological parameters, as well as their immune functions. OTA also modifies the gut microbiota of poultry, decreasing its richness and diversity [8,9,10,11].

T-2 mycotoxin is mainly produced by *Fusarium* species and belongs to the trichothecenes family. Since data about the occurrence and toxic effects of T-2 toxin are limited, there are indicative levels of the European Commission for the sum of T-2 and HT-2, above which research is needed [4].

In broiler chickens, the toxic effects of T-2 toxin can be manifested by the impairment of performance and immunocompetence, oral lesions, alteration in blood parameters and organ weights, and induce apoptosis in chicken hepatocytes [12,13,14,15].

OTA and T-2 toxin combination resulted in additive adverse effects, causing a decrease in body weight and feed intake [16,17], depressed serum concentrations of total protein, lactate dehydrogenase activity [16], and an impairment of immune function in broiler chickens [18,19].

In this context, the use of adequate detoxification programs is highly recommended. The most well-known approach for the prevention of mycotoxins involves the use of inert adsorbents to bind mycotoxins, reducing their bioavailability in the gastrointestinal tract of animals [20,21,22]. 

A novel multicomponent mycotoxin detoxifying agent (MMDA) containing a modified zeolite (clinoptilolite), *Bacillus subtilis, Bacillus Licheniformis, Saccharomyces cerevisiae* cell wall, and silymarin has been developed, which presents adsorption, biotransformation, hepatoprotection, and immunostimulation as modes of action. 

Therefore, the objective of this study was to evaluate the efficacy of the mycotoxin detoxifying feed additive (MMDA or MycoRaid) (Patent Co., Mišićevo, Serbia) [23] as a detoxifier of OTA and T-2 toxin on broilers.

## 2. Materials and Methods

### 2.1. Broilers, Diets, and Experimental Design

A total of 240 1-day-old male chicks (Ross 308) were randomly allotted to 60 battery cages with an available surface of 0.305 m^2^ and 37 cm in height. 

The standard lighting program was 24 h of light for 2 days, 18 h of light and 6 h of dark per day until 7 days, and 16 h of light per day and 8 h of dark, thereafter. The temperature program was adjusted as follows: 0–2 days: 32–34 °C; 3–7 days: 29–31 °C; 2nd week: 26–28 °C; 3rd week: 23–25 °C; 4th week: 20–22 °C and 19–21 °C afterwards. Chickens were fed starter diets from 1 to 10 days, grower diets from 11 to 21 days, and finisher diets from 22 to 35 days. Basal diets were based on maize, wheat, and soybean meal, and were formulated according the nutrient requirements for Ross 308 [24] strain broilers. The composition and the estimated nutrient content of basal experimental diets are presented in Table 1. Feed and water were provided ad libitum, in an individual frontal linear feeder and two nipple drinkers connected to each cage. There were five dietary treatments, replicated 12 times each with 4 broiler chickens per replicate at the beginning of the experimental period. Broiler chickens were assigned to one of the five treatments of: (1) control (non-contaminated diet); (2) non-contaminated diet + 3 g/kg MMDA; (3) non-contaminated diet + 0.5 mg/kg OTA + 1 mg/kg T-2 toxin; (4) non-contaminated diet + 0.5 mg/kg OTA + 1 mg/kg T-2 toxin + 1 g/kg MMDA; and (5) non-contaminated diet + 0.5 mg/kg OTA + 1 mg/kg T-2 toxin + 3 g/kg MMDA. MMDA mycotoxin binder was provided by Patent Co. (Subotica, Serbia).

Ochratoxin A was produced by infecting sterile corn with *Aspergillus ochraceus*. The infected corn was incubated at 25 °C for 15 days. T-2/HT-2 was produced by infecting sterile corn with *Fusarium langsethiae* Fe 2391. The infected corn was incubated at 10 °C for 10 days. The contaminated substrate was then dried on 105 °C for 2 days, ground, and analyzed using LC-MS/MS.

Before starting the feed production, the three main raw ingredients used were analyzed by Patent Co. (Subotica, Serbia) for the content of the mycotoxins aflatoxins, deoxynivalenol (DON), zearalenone (ZEN), fumonisins (FBs), OTA, T-2 y HT-2 toxins. Soybean meal and wheat presented values for all analyzed mycotoxin below their quantification limits. The maize contained only low contents of Fumonisin B_1_ (0.64 mg/kg) and Fumonisin B_2_ (0.22 mg/kg). These ingredients did not contain the mycotoxins under evaluation OTA, or T-2 toxin, and were considered suitable for experimental feeds preparation. Contaminated diets were prepared by adding the culture powder containing OTA and T2 toxin at a level of 1.41% to the diet to obtain the levels of 0.5 mg/kg of OTA and 1 mg/kg of T2 toxin. Experimental diets were analyzed by HPLC for OTA and T-2 toxin, aflatoxins, DON, ZEN, HT-2 toxin, and FBs at Patent Co. (Subotica, Serbia). 

### 2.2. Productive Parameters

Chickens were bulk weighed at day 1 and weighed per cage at 10, 21, and 35 days. The body weight (BW), body weight gain (BWG), feed intake, and feed to gain ratio in g feed/g gain (FCR) were calculated for the periods 1–10, 10–21, and 21–35 days, and the overall study. Mortality was checked and recorded daily.

### 2.3. Relative Weight of Organs, Breast Meat Yield, and Oral Lesions

On day 36, a total of 120 animals, 24 chickens per treatment (2 from each replicate or cage), were randomly selected, tagged, and euthanized. Breast, gizzard, liver, kidneys, and spleen were collected and weighed. Breast meat yield and relative organ weights were expressed as a percentage of BW. The oral lesions were assessed using the following score: grade 0 refers to no lesions; grade 1 refers to color of the tongue ash or black; grade 2 refers to lesions with white–yellow plaques; lesions outside buccal cavity (beak, crest, wattle, and eyelids); grade 3 refers to necrotic points, lesions inside buccal cavity (buccal cavity, tongue, tongue papillae, pores of saliva glands); and grade 4 refers to lesions in and outside of cavity.

### 2.4. Blood Hematology and Biochemistry

At day 36, blood samples from one chicken per replicate (2 mL/chicken) were collected by cardiac puncture into non-heparinized tubes for blood hematology and serum biochemistry. The hematological traits were determined using a CELL-DYN 3700 hematology analyzer (Abbott, Chicago, IL, USA), as detailed in Riahi et al. [25]. The heterophil to lymphocyte ratio was evaluated as a measure of stress. 

Serum samples were obtained by centrifugation at 1000× *g* for 10 min. Total protein, albumin, triglycerides, cholesterol, glucose, aspartate aminotransferase (AST), alanine aminotransferase (ALT), gamma-glutamyl transferase (GGT), and lactate dehydrogenase (LDH) were measured by an automatic biochemical analyzer (Olympus AU5800, Beckman Coulter, Brea, CA, USA).

### 2.5. Intestinal pH and Morphometry

The pH of the gastrointestinal content (2 chickens/pen) was measured with a portable pH-meter Hanna model HI-9125 (Limena, Italy) and pH electrode Crison model 52-32 (Barcelona, Spain), calibrated before the measurements with buffer solutions at pH 4 and 7. The pH of the jejunum content was measured following two procedures: in situ, directly in content of the distal part of jejunum, after insertion of a pH probe directly into the gut lumen post euthanasia [26], and ex situ, where jejunum digesta was removed and 1/10 dilution with deionized water was performed prior to pH determination [27].

Intestinal section samples (jejunum) were fixed in a solution of 10% neutral-buffered formalin at 4 °C. Tissue was sectioned at 4 μm thickness (three cross-sections from each sample) and stained with haematoxylin and eosin. Morphometric measurements were performed with a light microscope (BHS, Olympus, Barcelona, Spain) as described by Nofrarias et al. [28]. The villus height (VH) and crypt depth (CD) from each segment of each chicken were examined on a linear ocular micrometer (Olympus, 209-35040, Micro Planet, Barcelona, Spain) with a Leica Camera DFC320 (Leica Microsystems Ltd., Wetzlar, Germany) coupled to a computer-based image analysis system LAS v.3.8. (Leica Microsystems Ltd., Wetzlar, Germany). Villus:crypt ratio was calculated by dividing villus height by crypt depth.

### 2.6. Statistical Methods

The basic study design was a randomized complete block design (RCB) of 5 dietary treatments allocated in 12 blocks, with cage location as block criteria. Data were subjected to an analysis of variance to examine the main effect of dietary treatment using the General Linear Model (GLM) procedure of SAS System for Windows V.9.4 (SAS Institute, Cary, NC, USA, 2019). Significant differences were declared at *p* ≤ 0.05, while *p* values between 0.05 and 0.10 were considered a near-significant trend. Tukey’s adjustment was applied to perform multiple comparisons of means in case the F value for treatment in the ANOVA table were significant. In addition, the following linear contrasts were tested: Mycotoxin presence (non-contaminated vs. contaminated): T-1, T-2 vs. T-3, T-4, T-5. Mycotoxin binder at 3 g/kg feed: T-1, T-3 vs. T-2, T-5. Binder on contaminated diets: T-3 vs. T-4, T-5.

## 3. Results

### 3.1. Analyzed Nutrients and Dietary Mycotoxin Concentrations

Nutrients and mycotoxins present in the experimental diets are shown in Table 2. The analyzed total protein content of grower and finisher diets was around 0.5% higher than expected values according to feed formulation: 22.08% and 20.08% versus 21.5% and 19.5%, respectively. Regarding mycotoxins, no aflatoxins (<limit of quantification (LOQ) = 0.4 µg/kg), DON (<LOQ = 64 µg/kg) or ZEN (<LOQ = 16 µg/kg) were found in any of the experimental diets. As expected, according to the level of FBs found in the maize used for feed preparation, these mycotoxins were present in all the feeds (FB1 between 131 and 276 µg/kg; FB2 between <40 to 88 µg/kg), but at levels far below those recommended by the European Commission [4]. The analysis of feeds confirmed the presence of OTA in all diets of treatments T-3 to T-5. The expected value was 500 µg/kg and the average analyzed values were 718, 536, and 426 µg/kg for starter, grower, and finisher feeds, respectively. OTA presence in feeds of treatments T-1 and T-2 was in all cases lower than the LOQ (<1.6 µg/kg), confirming that no mycotoxin cross-contamination was produced. The results of T-2 toxin in feeds followed the same trend than those observed for OTA. Feeds from treatments T-1 and T-2 presented values of T-2 toxin below the LOQ (<9.6 µg/kg). The expected content of T-2 toxin in experimental feeds from treatments T-3 to T-5 was 1000 µg/kg, and the analyzed values were on average 853, 990, and 1025 µg/kg for starter, grower, and finisher feeds, respectively.

### 3.2. Productive Parameters

The results of productive parameters and mortality are shown in Table 3. In the starter phase (from 1 to 10 days), significant statistical differences between treatments were observed in all the productive parameters evaluated. The linear contrast “mycotoxin” confirmed that broilers fed diets co-contaminated with OTA and T-2 toxin presented lower feed intake (*p =* 0.0003) than chickens fed non-contaminated feeds, and their growth (*p* = 0.0002), and final BW were also statistically lower (*p =* 0.0002). The BW of broilers at 10-d fed contaminated diets was 7.2% lower in relation to chickens fed non-contaminated feeds (245.3 vs. 264.2 g). No effects were observed on broiler performance due to the inclusion of the feed additive MMDA mycotoxin binder, at the level of 1% or 3% (Table 3). The same pattern was observed in performance of broilers during the grower phase, from 10 to 21 days (Table 3). On average, the BW of chickens fed contaminated diets was 6.2% lower compared to broilers fed non-contaminated feeds (783 vs. 835 g) (linear contrast “Mycotoxin”, *p* = 0.0004). Statistically significant differences among treatments were obtained in this growing period in feed intake (*p* = 0.04), BWG (*p* = 0.01), and FCR (*p* = 0.03). Dietary treatment had a significant effect on productive parameters of broilers during the finisher period (from 21 to 35 days); feed intake (*p* = 0.01), BWG (*p* = 0.0009), and FCR (*p* = 0.0002). In this period, the feed additive MMDA mycotoxin binder included in OTA and T2 toxin-contaminated diets (at both 1 or 3 g/kg) significantly improved BWG (76.6 g/day vs. 82.3 g/day, *p* = 0.03) and FCR (1.59 vs. 1.54 g/g, *p =* 0.01). 

### 3.3. Blood Biochemistry

The results of biochemistry of blood of broilers at the end of study are presented in Table 4. Statistically significant differences among treatments were found on total proteins and albumin (*p* = 0.01). In the present study, the inclusion of 3 g of MMDA mycotoxin binder/kg feed seemed to revert the mycotoxin deleterious effects improving both values (*p* = 0.009, for total proteins and *p* = 0.04, for albumin). The additive at lower dose increased the triglycerides level (*p* = 0.0008) and at higher dose increased the activity of the LDH enzyme (*p* = 0.05). Statistical effects were also observed on the enzymatic AST activity (*p* = 0.04) related to the presence of the mycotoxins. 

### 3.4. Blood Hematology

The results of hematology of blood of broilers at the end of study are presented in Table 5 and Table 6. Dietary treatment had no effect on blood count of broilers (Table 5), except for mean corpuscular volume (MCV) (*p* = 0.04). The additive increased the hematocrit value of broilers fed contaminated diets (linear contrast “additive”, *p* = 0.02). The inclusion of increasing levels of the detoxifying product to the mycotoxin contaminated diet produced a progressive increase in the total number of red blood cells (RBC), which was statistically significant at the level of 3 g/kg (*p* = 0.02). A significant reduction in the number of monocytes was also observed as the level of the additive increased (*p* = 0.03). Other blood parameters, even heterophil to lymphocyte ratio (H/L) (related to stress) were not affected by dietary treatments.

### 3.5. Breast Meat Yield

Statistically significant differences among treatments in BW (*p* = 0.002) and breast weight (*p* = 0.0001), and the corresponding linear contrasts confirmed that these differences were mainly related to the dietary presence of mycotoxins (BW, *p* = 0.0008; breast meat weight, *p* = 0.0001). When broilers were fed mycotoxin-contaminated diets, the inclusion of the feed additive MMDA mycotoxin binder showed a tendency to improve BW (*p* = 0.08). At 36 days of life, the final BW of chickens fed contaminated diet and 1 g/kg was numerically 4.6% higher than the BW of chickens fed control contaminated feed (1995 vs. 1907 g). When the inclusion level of the MMDA mycotoxin binder was 3 g/kg, the observed body weight increase was 7.3% (2046 vs. 1907 g) (Table 7). A clear effect of the mycotoxins was observed in the percentage of breast meat that was higher (17.5%) for broilers fed non-contaminated diets compared to chickens that consumed feeds with OTA and T-2 toxin (15.7%) (*p* = 0.0001). 

### 3.6. Relative Weight of Organs and Oral Lesions

No effects of dietary treatments were observed in the relative weights of gizzard, kidney, or spleen organs, expressed in percentage of body weight, except a tendency in liver weight (*p* = 0.08) (Table 8). The number of chickens showing oral lesions is also reported in Table 8. Most of the chickens presented grade 0 (no lesions) and only six chickens presented grade 1, and no related to the dietary treatment tested. No animals presented grades 2, 3, or 4.

### 3.7. Intestinal Content pH and Intestinal Morphometry

The values of intestinal content pH and intestinal morphometry parameters are reported in Table 9. No statistically significant differences were found among treatments irrespective to the method used for intestinal pH measurement (in situ or in an aqueous dilution). No significant differences among treatments were observed for any of the villus height, crypt depth, and the ratio of villus height to crypt depth (*p* > 0.05).

## 4. Discussion

Due to their constant occurrence in livestock feeds and their toxicity in animals, including poultry species, the evaluation of the toxicity of OTA and T-2 toxin in broiler chickens feed and its detoxification, by using a detoxifying agent, is considered very important. In the current research, contaminated diet with OTA and T-2 toxin (T-3) reduced significantly the BWG of broilers, the feed intake and impaired the FCR. Similarly, García et al. [17], using the same doses as the present study, demonstrated that broiler chickens fed 0.567 mg/kg of OTA and 0.927 mg/kg of T-2 toxin had a lower body weight and feed intake reduction. The dose of OTA used here (0.5 mg/kg feed) was higher than the guidance value (0.1 mg/kg feed) set by the European Commission [4]. Therefore, this dose was high enough to cause detrimental effects in gain weight, feed intake and feed conversion ratio. The feeding of OTA (0.25 mg/kg feed) and T-2 toxin (0.5 mg/kg feed) decreased the body weights and feed consumption of chickens [16]. Kubena et al. [16] reported that broiler chickens fed OTA—T-2 diets (2 and 4 mg/kg feed) depressed the body weight of chickens and impaired the feed conversion ratio. The observed growth depression in mycotoxins-treated chicks may be due to the inhibition of protein synthesis followed by secondary disruption of RNA and DNA synthesis provoked by both mycotoxins [16]. Furthermore, this depression may be due to inflammation, contact erosion, and irritation of gastrointestinal tract, resulting into decrease in feed consumption and, consequently, in a decrease in body weight of chickens fed mycotoxins [29]. Interestingly, the toxin binder significantly counteracted the adverse effect of mycotoxins in performance. 

Performance results did not reach the goals stated by the Aviagen company for male broiler chicken Ross 308 [24] under good management and environmental conditions when feeding pelleted diets with adequate nutrient levels, probably due to the supply of feeds in mash form, and the allocation of the animals in cages.

Concentrations of blood proteins and albumin in serum were lower in chickens fed OTA and T-2 toxin compared to control chickens. As described by Kubena et al. [16] and Wang et al. [18], the concentrations of proteins and albumin significantly decreased by the combination of OTA and T-2 toxin. This agrees with the study of Pozzo et al. [10], in which the blood proteins and albumin of intoxicated chickens (0.1 mg OTA/kg feed) significantly decreased. Mnafi et al. [30] also reported a significant decrease in the blood proteins and albumin of chickens fed 0.5 mg/kg T-2 for 5 weeks. The results of the current study on the serum total proteins and albumin may reflect liver function, as OTA is known to impair hepatic protein synthesis [8]. In addition, the decrease in the blood proteins and albumin levels may be due to the degeneration of endoplasmic reticulum and the inhibition of protein synthesis in the hepatocytes, as T-2 toxin is a known inhibitor of eukaryotic protein synthesis by impairing initiation and termination steps of protein synthesis [16]. On the other hand, a possible increased protein elimination mechanism and renal leakage of albumin induced by OTA through the kidneys could be involved [9]. The inclusion of 3 g of MMDA mycotoxin binder/kg feed mitigated the deleterious effects on these biochemical parameters, suggesting the efficacy of this product in reverting the disorders of hepatic protein synthesis. 

Monocytes are a type of white blood cell produced in the bone marrow that enter the blood, and migrate to tissues (the spleen, liver, lung, and bone marrow), where they become macrophages. Monocytes are part of the innate immune system, and are related to the processes of intestinal inflammation, which in the case of chickens results in lower productivity. A significant reduction in the number of monocytes was also observed as the level of the additive increased. The inclusion of the additive in the mycotoxin contaminated feed, led to a reduction in the number of monocytes that could indicate less intestinal inflammation, which would represent a beneficial effect for chickens. It has been described that subcutaneous OTA exposure increased levels of monocytes in broiler chickens [31]. Other blood parameters were not affected by dietary treatments. Several researchers have reported that H/L can be used as a good hematological indicator of stress response in chickens [32]. In the present study, the mycotoxin-contaminated diet did not affect the H/L ratio, speculating that OTA and T-2 mycotoxins at levels used did not cause suppression of hematopoiesis in the bone marrow. Similar results were reported by Pozzo et al. [10], as most of the hematological parameters were not affected by an OTA contaminated diet.

The reduction of the breast meat observed in broilers fed contaminated diets with mycotoxins is a direct result of the reduction of broilers body weights. Additionally, this reduction may be due to the possible carry-over of these mycotoxins, particularly OTA, to the muscle, as a consequence of contaminated feed ingestion. Due to the carry-over from feed, OTA has been widely reported to occur in meat and meat by-products [3]. In chickens exposed to 0.5 mg of OTA/animal weekly for four weeks, OTA was found at 0.28 and 0.20 ng/g in breast and thigh muscle, respectively, after the first two weeks of exposure. After four weeks, OTA residue in muscle increased slightly, reaching its maximum value (0.84 ng/g in both white and red muscles) [2].

Moreover, it was reported that OTA or T-2 mycotoxin could affect the immune system by the atrophy of the lymphoid organs along with depletion of lymphocytes and the enlargement of the kidney and liver [9,13,31]. However, in the current research, the combination of OTA and T-2 mycotoxin at levels examined in broilers feed did not affect the relative weights of liver, gizzard, kidneys, and spleen (*p* > 0.05). It was reported that feeding broiler chickens 0.1 mg/kg OTA in diet did not affect relative weights of liver, kidney and bursa of Fabricius [33]. Furthermore, no significant differences in the relative weights of bursa of Fabricius and spleen were found in broilers exposed to 2 mg/kg of OTA and 3 mg/kg of T-2 toxin [29]. Therefore, it can be concluded that the reported effects of dietary OTA and T-2 toxin on relative organ weights in poultry are very inconsistent across studies. 

The occurrence and severity of the oral lesions appear to reflect the effect of T-2 toxin [16]. It has been observed in chickens who received 2 mg/kg T-2 toxin, leading to loss of body weight [34]. Nevertheless, under current experimental conditions, no oral lesions have been found. In terms of oral lesions, chickens may be relatively resistant to the effects of T-2 mycotoxin, when the concentration in the feed does not exceed 1 mg/kg. Moreover, it should be noted that the feeding of chickens with contaminated diets containing 2 mg of OTA/kg of feed did not produce any change in oral lesions scores [35].

The gastrointestinal tract is the site of nutrient digestion and absorption and is a target organ of mycotoxicosis. It has been found that several mycotoxins, including OTA and T-2 mycotoxin, result in histological changes in broilers intestines such as shortened, atrophied and thin villi, and elongated crypts with irregular forms, indicative that these mycotoxins can alter the digestive tract and the absorption function [36]. However, in the present study, the combination of OTA and T-2 mycotoxin at levels tested in broilers feed did not affect villus height and crypt depth. Similar results have been observed in broilers fed other mycotoxins, such as DON [37]. 

## 5. Conclusions

Based on the results of the present study, the ingestion of OTA and T-2 toxin adversely affected the productive parameters and some blood and carcass characteristics of broiler chickens. The addition of the MMDA detoxifying agent at 1 or 3 g/kg feed in contaminated diets improved performance, mainly in the finisher period (from 21–35 days), and was partly capable of counteracting the deleterious effects of these mycotoxins. In particular, the simultaneous presence of OTA and T2 toxin significantly reduced the BWG of broilers, the feed intake and impaired the FCR during the finished period. Broiler chickens fed mycotoxin contaminated feed showed lower serum concentrations of blood proteins and albumin were lower in chickens fed the mycotoxins compared to control chickens, which may reflect an impairment of hepatic protein synthesis. The inclusion of 3 g of MMDA mycotoxin binder/kg feed mitigated the deleterious effects on these biochemical parameters, suggesting the efficacy of this product in reverting the disorders of hepatic protein synthesis. 

## Figures and Tables

**Table 1 animals-11-03205-t001:** Formulation and proximate analysis of experimental basal diets.

Ingredients (%)	0–10 d	10–21 d	21–35 d
Maize	33.90	35.03	39.01
Wheat	20.00	20.00	20.00
Soybean meal (48% CP)	37.23	35.48	30.90
Soybean oil	4.52	5.80	6.70
DL-methionine	0.34	0.23	0.26
L-lysine-HCl	0.29	0.16	0.11
L-threonine	0.13	0.07	0.05
Calcium carbonate	1.10	0.96	0.84
Monocalcium phosphate	1.51	1.33	1.19
Sodium chloride	0.16	0.20	0.22
Sodium bicarbonate	0.32	0.25	0.23
Choline chloride	0.09	0.07	0.07
Vitamin-mineral premix ^1^	0.40	0.40	0.40
Noxyfeed ^2^	0.02	0.02	0.02
Nutrient content (%)			
AME (kcal/kg)	3000	3100	3200
Crude protein	22.5	21.50	19.50
Ash	6.33	5.90	5.41
Crude fiber	2.48	2.43	2.34
Crude fat	6.91	8.21	9.20
Digestible lysine	1.34	1.19	1.03
Digestible methionine	0.64	0.52	0.52
Digestible Met + Cys	0.95	0.82	0.80
Digestible threonine	0.87	0.78	0.69
Digestible valine	0.96	0.93	0.84
Digestible arginine	1.41	1.36	1.21
Digestible tryptophan	0.23	0.22	0.20
Total calcium	0.96	0.87	0.79
Total phosphorus	0.73	0.68	0.63
Non-phytate phosphorus	0.48	0.435	0.395

^1^ Vitamin–mineral premix provided the following nutrients per kg of diet: vitamin A, 13,500 IU; vitamin D3, 4800 IU: vitamin E, 67 IU; vitamin B1: 3 mg; vitamin B2, 9 mg; vitamin B6, 4.5 mg; vitamin B12, 16.5 µg; vitamin K3, 3 mg; calcium pantothenate, 16.5 mg; nicotinic acid, 51 mg; folic acid 1.8 mg, biotin: 30 µg; Fe, 54 mg; I, 1.2 mg; Co, 0.6 mg; Cu, 12 mg; Mn, 90 mg; Zn, 66 mg; Se, 0.18 mg; Mo, 1.2 mg; ^2^ Contains BHT+ propyl gallate (56%), and citric acid (14%) (ITPSA, Barcelona, Spain).

**Table 2 animals-11-03205-t002:** Analyzed nutrients (%) and mycotoxins (µg/kg) in experimental diets.

Item	0–10 d (Starter)	10–21 d (Grower)	21–35 d (Finisher)
T-1 (analyzed nutrients, %)			
Dry matter	88.08	88.5	89.43
Crude protein	22.94	22.16	20.46
Ether extract	5.75	7.82	8.91
Ash	5.47	5.93	4.99
T-1 (analyzed mycotoxins, µg/kg)			
OTA ^1^	<1.6	<1.6	<1.6
FB1	154.5	201.5	263.9
FB2	<40	40.10	42.16
HT-2 toxin	<9.6	<9.6	<9.6
T-2 toxin	<9.6	<9.6	<9.6
T-2 (analyzed nutrients, %)			
Dry matter	88.43	88.64	89.76
Crude protein	23.02	22.31	19.78
Ether extract	6.29	7.39	8.84
Ash	5.82	5.77	5.27
T-2 (analyzed mycotoxins, µg/kg)			
OTA	<1.6	<1.6	<1.6
FB1	150.0	190.3	259.6
FB2	<40	<40	50.3
HT-2 toxin	<9.6	<9.6	<9.6
T-2 toxin	<9.6	<9.6	<9.6
T-3 (analyzed nutrients, %)			
Dry matter	88.61	88.57	89.58
Crude protein	22.45	21.93	19.90
Ether extract	5.99	7.52	8.72
Ash	5.69	5.50	4.92
T-3 (analyzed mycotoxins, µg/kg)			
OTA	685.3	535.4	436.1
FB1	153.2	276.5	258.8
FB2	<40	79.7	88.1
HT-2 toxin	254.7	267.9	300.7
T-2 toxin	827.7	951.7	985.6
T-4 (analyzed nutrients, %)			
Dry matter	88.48	88.79	89.62
Crude protein	22.36	21.84	20.12
Ether extract	5.80	7.45	8.72
Ash	5.88	5.44	5.00
T-4 (analyzed mycotoxins, µg/kg)			
OTA	749.1	600.7	441.3
FB1	133.3	175.6	238.9
FB2	<40	<40	68.45
HT-2 toxin	249.2	273.7	310.4
T-2 toxin	831.4	939.8	998.5
T-5 (analyzed nutrients, %)			
Dry matter	88.73	89.10	89.32
Crude protein	22.53	22.15	20.16
Ether extract	5.82	7.40	8.53
Ash	5.92	5.66	5.23
T-5 (analyzed mycotoxins, µg/kg)			
OTA	719.5	471.7	400.3
FB1	131.3	191.9	247..2
FB2	<40	52.2	62.3
HT-2 toxin	256.4	311.4	334.8
T-2 toxin	899.6	1080	1090

^1^ OTA, ochratoxin A; FB, fumonisin.

**Table 3 animals-11-03205-t003:** Effects of feeding ochratoxin A (OTA) and T-2 mycotoxins and the addition of a mycotoxin binder product (MMDA) on broiler chickens performance.

Treatment	Mycotoxins (mg/kg)	MMDA ^1^ (g/kg)	BW ^2^	BWG ^3^	Feed Intake	FCR ^4^
Starter						
T-1	Non-contaminated	0	267.8 ^a^	22.4 ^a^	27.1 ^a^	1.22 ^b^
T-2	Non-contaminated	3	260.6 ^ab^	21.7 ^ab^	27.1 ^a^	1.25 ^ab^
T-3	0.5 OTA ^5^ + 1 T-2	0	248.4 ^ab^	20.4 ^ab^	25.7 ^b^	1.26 ^ab^
T-4	0.5 OTA + 1 T-2	1	242.0 ^b^	19.8 ^b^	25.5 ^b^	1.30 ^a^
T-5	0.5 OTA + 1 T-2	3	245.4 ^b^	20.1 ^b^	25.5 ^b^	1.29 ^ab^
SEM ^6^			17.74	1.77	1.48	0.06
*p*-Value			0.003	0.003	0.008	0.02
Linear contrast						
“Mycotoxin”: T-1, T-2 vs. T-3, T-4, T-5			0.0002	0.0002	0.0003	0.006
“3 g/kg additive”: T-1, T-3 vs. T-2, T-5			0.32	0.32	0.76	0.09
“Additive”: T-3 vs. T-4, T-5			0.46	0.46	0.64	0.15
Grower						
T-1	Non-contaminated	0	848 ^a^	52.7 ^a^	71.5 ^a^	1.35 ^b^
T-2	Non-contaminated	3	822 ^ab^	51.0 ^ab^	70.8 ^ab^	1.39 ^ab^
T-3	0.5 OTA + 1 T-2	0	783 ^b^	48.6 ^ab^	67.6 ^bc^	1.41 ^a^
T-4	0.5 OTA + 1 T-2	1	770 ^b^	48.0 ^b^	67.1 ^c^	1.40 ^ab^
T-5	0.5 OTA + 1 T-2	3	796 ^ab^	49.8 ^ab^	69.4 ^abc^	1.40 ^ab^
SEM			52.1	3.56	4.04	0.04
*p*-Value			0.02	0.01	0.04	0.03
Linear contrast						
“Mycotoxin”: T-1, T-2 vs. T-3, T-4, T-5			0.0004	0.002	0.006	0.006
“3 g/kg additive”: T-1, T-3 vs. T-2, T-5			0.65	0.80	0.64	0.29
“Additive”: T-3 vs. T-4, T-5			0.98	0.82	0.65	0.68
Finisher						
T-1	Non-contaminated	0	2093 ^a^	88.9 ^a^	134 ^a^	1.50 ^c^
T-2	Non-contaminated	3	2044 ^ab^	87.3 ^a^	133 ^a^	1.53 ^bc^
T-3	0.5 OTA + 1 T-2	0	1855 ^c^	76.6 ^b^	123 ^b^	1.59 ^a^
T-4	0.5 OTA + 1 T-2	1	1929 ^bc^	82.1 ^ab^	128 ^ab^	1.56 ^ab^
T-5	0.5 OTA + 1 T-2	3	1939 ^bc^	82.5 ^ab^	126 ^ab^	1.53 ^bc^
SEM			129.2	7.25	8.89	0.041
*p*-Value			0.0002	0.0009	0.01	0.0002
Linear contrast						
“Mycotoxin”: T-1, T-2 vs. T-3, T-4, T-5			0.0001	0.0002	0.0009	0.0004
“3 g/kg additive”: T-1, T-3 vs. T-2, T-5			0.65	0.31	0.59	0.24
“Additive”: T-3 vs. T-4, T-5			0.10	0.03	0.18	0.01

^1^ MMDA, mycotoxin binder; ^2^ BW, body weight; ^3^ BWG, body weight gain; ^4^ FCR, feed conversion ratio; ^5^ OTA, ochratoxin A; ^6^ SEM, standard error of mean (*n* = 12); ^a,b,c^: means values with different superscripts within the same column differ (*p* ≤ 0.05).

**Table 4 animals-11-03205-t004:** The effects of feeding ochratoxin A (OTA) and T-2 mycotoxins and the addition of a binder mycotoxin product (MMDA) on blood biochemistry of broiler chickens.

Treatment	Mycotoxins (mg/kg)	MMDA ^1^ (g/kg)	LDH (U/L) ^2^	GGT (U/L) ^3^	AST (U/L) ^4^	ALT (U/L) ^5^	ALB (g/L) ^6^	Tot prot (g/L) ^7^	Trig (mg/dL) ^8^	Chol (mg/dL) ^9^	Glu (mg/dL) ^10^
T-1	Non-contaminated	0	1893	23.3	358 ^a^	2.42	10.5 ^ab^	28.8 ^ab^	76.4 ^b^	129	252
T-2	Non-contaminated	3	1838	25.7	316 ^ab^	2.00	11.1 ^a^	30.7 ^a^	76.8 ^b^	132	249
T-3	0.5 OTA ^11^ + 1 T-2	0	1407	23.5	305 ^ab^	2.08	9.83 ^b^	27.2 ^b^	71.7 ^b^	126	253
T-4	0.5 OTA + 1 T-2	1	1727	21.8	280 ^ab^	2.33	10.4 ^ab^	28.6 ^ab^	112 ^a^	136	259
T-5	0.5 OTA + 1 T-2	3	2965	25.2	254 ^b^	2.08	10.2 ^ab^	28.2 ^b^	81.2 ^b^	129	252
SEM ^12^			1315	5.34	73.6	0.54	0.81	1.92	23.9	12.0	14.3
*p*-Value			0.07	0.43	0.04	0.28	0.01	0.01	0.0008	0.30	0.61
Linear contrast											
“Mycotoxin”: T-1, T-2 vs. T-3, T-4, T-5			0.69	0.51	0.007	0.77	0.005	0.001	0.07	0.98	0.28
“3 g/kg additive”: T-1, T-3 vs. T-2, T-5			0.06	0.20	0.07	0.19	0.04	0.009	0.48	0.40	0.63
“Additive”: T-3 vs. T-4, T-5			0.05	0.98	0.17	0.52	0.08	0.07	0.004	0.12	0.69

^1^ MMDA, mycotoxin binder; ^2^ LDG, lactate dehydrogenase; ^3^ GGT, gamma-glutamyl transferase; ^4^ AST, aspartate aminotransferase; ^5^ ALT, alanine aminotransferase; ^6^ ALB, albumin; ^7^ Tot prot, total proteins; ^8^ Trig, triglycerides; ^9^ Chol, cholesterol; ^10^ Glu, glucose; ^11^ OTA, ochratoxin A; ^12^ SEM, standard error of mean (*n* = 12); ^a,b^: means values with different superscripts with the same column differ (*p* ≤ 0.05).

**Table 5 animals-11-03205-t005:** The effects of feeding ochratoxin A (OTA) and T-2 mycotoxins and the addition of a binder mycotoxin product (MMDA) on blood hematology (blood count) of broiler chickens.

Treatment	Mycotoxins (mg/kg)	MMDA ^1^ (g/kg)	HCT ^2^	Hgb ^3^	RBC ^4^	MCV ^5^	MCH ^6^	MCHC ^7^
T-1	Non-contaminated	0	30.1	13.0	2.54	118 ^ab^	51.2	43.6
T-2	Non-contaminated	3	29.7	13.5	2.69	112 ^b^	50.0	44.6
T-3	0.5 OTA ^8^ + 1 T-2	0	28.2	12.4	2.50	115 ^ab^	49.6	43.0
T-4	0.5 OTA + 1 T-2	1	31.6	12.8	2.54	124 ^a^	50.4	40.8
T-5	0.5 OTA + 1 T-2	3	30.1	13.0	2.59	116 ^ab^	50.1	43.4
SEM ^9^			2.87	0.97	0.18	9.32	1.53	3.65
*p*-Value			0.11	0.17	0.13	0.04	0.15	0.14
Linear contrast								
“Mycotoxin”: T-1, T-2 vs. T-3, T-4, T-5			0.86	0.08	0.15	0.21	0.24	0.08
“3 g/kg additive”: T-1, T-3 vs. T-2, T-5			0.43	0.07	0.02	0.34	0.42	0.49
“Additive”: T-3 vs. T-4, T-5			0.02	0.16	0.31	0.16	0.23	0.46

^1^ MMDA, mycotoxin binder; ^2^ HCT, hematocrit; ^3^ Hgb, hemoglobin; ^4^ RBC, red blood cells; ^5^ MCV, mean corpuscular volume; ^6^ MCH, mean cell hemoglobin; ^7^ MCHC, mean corpuscular hemoglobin concentration; ^8^ OTA, ochratoxin A; ^9^ SEM, standard error of mean (*n* = 12); ^a,b^: means values with different superscripts with the same column differ (*p* ≤ 0.05).

**Table 6 animals-11-03205-t006:** The effects of feeding ochratoxin A (OTA) and T-2 mycotoxins and the addition of a binder mycotoxin product (MMDA) on blood hematology (leukogram) of broiler chickens.

Treatment	Mycotoxins (mg/kg)	MMDA ^1^ (g/kg)	Leuk ^2^	EOS ^3^ %	BAS ^4^ %	LYM ^5^ %	MON ^6^ %	HET ^7^ %	H/L ^8^
T-1	Non-contaminated	0	2.28	4.42	7.75	44.4	5.58 ^a^	37.8	0.83
T-2	Non-contaminated	3	1.90	4.64	7.09	46.0	3.36 ^ab^	38.9	1.12
T-3	0.5 OTA ^9^ + 1 T-2	0	2.07	6.42	4.90	40.2	4.50 ^ab^	45.4	1.20
T-4	0.5 OTA + 1 T-2	1	1.84	5.83	5.42	43.5	2.58 ^b^	42.1	1.12
T-5	0.5 OTA + 1 T-2	3	2.07	4.25	5.25	44.8	2.82 ^ab^	42.2	0.99
SEM ^10^			0.06	4.02	3.96	11.5	2.47	14.8	0.66
*p*-Value			0.52	0.82	0.41	0.85	0.03	0.78	0.76
Linear contrast									
“Mycotoxin”: T-1, T-2 vs. T-3, T-4, T-5			0.61	0.44	0.07	0.59	0.06	0.30	0.66
“3 g/kg additive”: T-1, T-3 vs. T-2, T-5			0.31	0.43	0.54	0.45	0.03	0.85	0.84
“Additive”: T-3 vs. T-4, T-5			0.64	0.58	0.73	0.31	0.05	0.47	0.51

^1^ MMDA, mycotoxin binder; ^2^ Leuk, leukocytes; ^3^ EOS, eosinophils; ^4^ BAS, basophils; ^5^ LYM, lymphocytes; ^6^ MON, monocytes; ^7^ HET, heterophils; ^8^ H/L, heterophils to lymphocytes ratio; ^9^ OTA, ochratoxin A; ^10^ SEM, standard error of mean (*n* = 12); ^a,b^: means values with different superscripts with the same column differ (*p* ≤ 0.05).

**Table 7 animals-11-03205-t007:** The effects of feeding ochratoxin A (OTA) and T-2 mycotoxins, and the addition of a binder mycotoxin product (MMDA) on breast meat yield of broiler chickens.

Treatment	Mycotoxins (mg/kg)	MMDA ^1^ (g/kg)	BW ^2^ (g)	Breast Meat Weight (g)	Breast Meat Yield (%)
T-1	Non-contaminated	0	2199 ^a^	385.8 ^a^	17.5 ^a^
T-2	Non-contaminated	3	2096 ^ab^	350.1 ^ab^	16.7 ^ab^
T-3	0.5 OTA ^3^ + 1 T-2	0	1907 ^b^	296.1 ^c^	15.7 ^b^
T-4	0.5 OTA + 1 T-2	1	1995 ^ab^	317.1 ^bc^	15.8 ^b^
T-5	0.5 OTA + 1 T-2	3	2046 ^ab^	322.4 ^bc^	15.7 ^b^
SEM ^4^			255.8	56.32	1.78
*p*-Value			0.002	0.0001	0.0009
Linear contrast					
“Mycotoxin”: T-1, T-2 vs. T-3, T-4, T-5			0.0008	0.0001	0.0001
“3 g/kg additive”: T-1, T-3 vs. T-2, T-5			0.73	0.69	0.29
“Additive”: T-3 vs. T-4, T-5			0.08	0.09	0.77

^1^ MMDA, mycotoxin binder; ^2^ BW, body weight; ^3^ OTA, ochratoxin A; ^4^ SEM, standard error of mean (*n* = 12); ^a,b,c^: means values with different superscripts with the same column differ (*p* ≤ 0.05).

**Table 8 animals-11-03205-t008:** The effects of feeding ochratoxin A (OTA) and T-2 mycotoxins and the addition of a binder mycotoxin product (MMDA) on relative weight of organs and oral lesions in broiler chickens.

Treatment	Mycotoxins (mg/kg)	MMDA ^1^ (g/kg)	Liver RW ^2^ (%)	Gizzard RW (%)	Kidneys RW (%)	Spleen RW (%)	Oral Lesions Scores (nº Chickens with Score = 1) ^3^
T-1	Non-contaminated	0	2.12	1.70	0.59	0.09	1 (of 24)
T-2	Non-contaminated	3	2.34	1.81	0.62	0.11	2 (of 24)
T-3	0.5 OTA ^4^ + 1 T-2	0	2.25	1.87	0.65	0.10	1 (of 24)
T-4	0.5 OTA + 1 T-2	1	2.25	1.83	0.60	0.10	0 (of 24)
T-5	0.5 OTA + 1 T-2	3	2.18	1.83	0.62	0.09	2 (of 24)
SEM ^5^			0.27	0.24	0.09	0.03	-
*p*-Value			0.08	0.16	0.12	0.32	-
Linear contrast							
“Mycotoxin”: T-1, T-2 vs. T-3, T-4, T-5			0.98	0.06	0.27	0.34	-
“3 g/kg additive”: T-1, T-3 vs. T-2, T-5			0.19	0.39	0.77	0.40	-
“Additive”: T-3 vs. T-4, T-5			0.62	0.59	0.07	0.90	-

^1^ MMDA, mycotoxin binder; ^2^ RW, relative weight; ^3^ scoring used for oral lesions evaluation: Grade 0—without lesions. Grade 1—color of the tongue ash or black. Grade 2—lesions with white-yellow plaques; lesions outside buccal cavity (beak, crest, wattle, and eyelids). Grade 3—necrotic points, lesions inside buccal cavity (buccal cavity, tongue, tongue papillae, pores of saliva glands). Grade 4—lesions in and outside of cavity; ^4^ OTA, ochratoxin A; ^5^ SEM, standard error of mean (*n* = 12).

**Table 9 animals-11-03205-t009:** The effects of feeding ochratoxin A (OTA) and T-2 mycotoxins and the addition of a binder mycotoxin product (MMDA) on intestinal pH and morphometry in broiler chickens.

Treatment	Mycotoxins (mg/kg)	MMDA ^1^ (g/kg)	Intestinal pH ^2^	Intestinal pH ^3^	Villus Height (µm)	Crypt Depth (µm)	Villus Height/Crypt Depth
T-1	Non-contaminated	0	6.46	7.01	928	128	7.76
T-2	Non-contaminated	3	6.30	6.90	959	124	7.96
T-3	0.5 OTA ^4^ + 1 T-2	0	6.48	6.93	913	123	7.59
T-4	0.5 OTA + 1 T-2	1	6.36	6.89	918	123	7.75
T-5	0.5 OTA + 1 T-2	3	6.53	7.00	937	128	7.52
SEM ^5^			0.51	0.43	32.6	5.8	0.289
*p*-Value			0.50	0.81	0.87	0.94	0.85
Linear contrast							
“Mycotoxin”: T-1, T-2 vs. T-3, T-4, T-5			0.41	0.81	0.49	0.80	0.38
“3 g/kg additive”: T-1, T-3 vs. T-2, T-5			0.58	0.83	0.41	0.93	0.83
“Additive”: T-3 vs. T-4, T-5			0.75	0.91	0.72	0.65	0.90

^1^ MMDA, mycotoxin binder; ^2^ intestinal pH measured directly in distal part of jejunum; ^3^ intestinal pH measured on a 1/10 dilution of jejunum content; ^4^ OTA, ochratoxin A; ^5^ SEM, standard error of mean (*n* = 12);

## Data Availability

All data sets collected and analyzed during the current study are available from the corresponding author upon fair request.

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
