# Peer review of "Effect of a Mycotoxin Binder (MMDA) on the Growth Performance, Blood and Carcass Characteristics of Broilers Fed Ochratoxin A and T-2 Mycotoxin Contaminated Diets"

_animals, 2021, doi:10.3390/ani11113205_

Round 1

Reviewer 1 Report

1.The introduction lacks background description to the MMDA and needs to be supplemented

2.Some key data can be displayed in the form of pictures

3.What is the detoxification mechanism of mycotoxin binder?Is it targeted binding or something else? Do you need a display of molecular docking?

Author Response

Responses to reviewer 1 comments:

First of all, thank you for your time,  for your revision ad comments.

Point 1. The introduction lacks background description to the MMDA and needs to be supplemented

Response 1: The description of the MMDA product tested has been included in Introduction section

“A novel multicomponent mycotoxin detoxifying agent (MMDA) containing a modified zeolite (Clinoptilolite), Bacillus subtilis, Bacillus Licheniformis, Saccharomyces cerevisiae cell wall and silymarin has been developed, which presents adsorption, biotransformation, hepatoprotection and immunostimulation as modes of action”.

Point 2. Some key data can be displayed in the form of pictures.

Response 2: We try to show in Tables all data obtained, to facilitate the understandig of the study, but if you could specify the data  you propose in Figures, we are open to perform the required changes.

Point 3. What is the detoxification mechanism of mycotoxin binder?Is it targeted binding or something else? Do you need a display of molecular docking?

Response 3: The product has adsorption, biotransformation, hepatoprotection and immunostimulation as mode of action. It binds AFB1, OTA, FB1, ZEN, T-2, ergot alkaloids and other toxins. The product has biotransformation activity against ZEN.

Reviewer 2 Report

In the present paper I carefully reviewed, the Authors have described the effects of a mycotoxin binder on the growth performance, blood and carcass characteristics of broilers fed ochratoxin A and T-2 mycotoxin contaminated diets..

I would like to congratulate Authors for the good-quality of their article, the literature reported used to write the paper, and for the clear and appropriate structure.

The manuscript is well written, presented and discussed, and understandable to a specialist readership.

In general, the organization and the structure of the article are satisfactory and in agreement with the journal instructions for authors. The subject is adequate with the overall journal scope.

The work shows a conscientious study in which a very exhaustive discussion of the literature available has been carried out.

The Introduction section provides sufficient background (however, additional references of recently published papers may add value to this section), and the other sections include results clearly presented and analyzed exhaustively.

However, as specific comments, with the aim to further improve the quality of the paper, the Conclusion section could be improved; also, the Authors have to check if alle references have been cited in the text.

Further, add the appropriate references related to the methods used.

Further, check the style of the references list according to the journal's guidelines. 

So, I recommend the acceptance of the paper after revision.

Author Response

Responses to reviewer 2 comments:

Point 1: General remarks: In the present paper I carefully reviewed, the Authors have described the effects of a mycotoxin binder on the growth performance, blood and carcass characteristics of broilers fed ochratoxin A and T-2 mycotoxin contaminated diets..

I would like to congratulate Authors for the good-quality of their article, the literature reported used to write the paper, and for the clear and appropriate structure.

The manuscript is well written, presented and discussed, and understandable to a specialist readership.

In general, the organization and the structure of the article are satisfactory and in agreement with the journal instructions for authors. The subject is adequate with the overall journal scope.

The work shows a conscientious study in which a very exhaustive discussion of the literature available has been carried out.

Response 1The authors thank very much the reviewer for his/her time to revise our manuscript.

Point 2: Specific remarks: The Introduction section provides sufficient background (however, additional references of recently published papers may add value to this section), and the other sections include results clearly presented and analyzed exhaustively.

However, as specific comments, with the aim to further improve the quality of the paper, the Conclusion section could be improved; also, the Authors have to check if alle references have been cited in the text.

Further, add the appropriate references related to the methods used.

Further, check the style of the references list according to the journal's guidelines. 

Response 2: We are very thankful again for giving us the opportunity to further improve the quality of our manuscript. We have performed the changes suggested by the reviewers with the aim to clearly present our research.

Some new studies have been examined and included in the Introduction section.

All referenced listed have been checked to be used in the text.

The references list has been written according to the journal’s guidelines.

Reviewer 3 Report

Line 17, 33, 36, 85: change “birds” by broilers

Line 18: write better what the experiment consists of. How are animals exposed to mycotoxins? indicate the groups there are ??

Line 22: change birds

Line 26: According to the writing, the detoxifier carries the mycotoxins themselves ????

Line 34: What is the contaminated control?

Line 115: But Soybean meal and wheat presented values for OTA and T2. Which???

Line 145, 147, 262, 297, 341, 347, 348, 353: change bird by chicken

Line 257-259: Both paragraphs indicate the same. UNIFY

Line 261: Indicate in the material and methods that H / L is used to study stress

Line 282-285: Reflect the data in the table, or change the table type, because what is indicated is not understood.

Line 286: its 17,5%

Line 296: delete in grams, because its not in the table

Author Response

Responses to reviewer 3 comments:

First of all, thank you very much for your revision and comments, that could improve the manuscript.

Point 1: Line 17, 33, 36, 85: change “birds” by broilers

Response 1: Checked and done.

Point 2: Line 18: write better what the experiment consists of. How are animals exposed to mycotoxins? indicate the groups there are ??

Response 2: Line 18-24:  In the present experiment, broiler chickens were allotted into 5 groups; group (1) received a non-contaminated diet, group (2) received non-contaminated diet + 3 g/kg of a mycotoxin binder (MMDA), group (3) received non-contaminated diet + 0.5 mg/kg OTA + 1 mg/kg T-2 toxin, group (4) received non-contaminated diet + 0.5 mg/kg OTA + 1 mg/kg T-2 toxin + 1 g/kg MMDA, and group (5) received non-contaminated diet + 0.5 mg/kg OTA + 1 mg/kg T-2 toxin + 3 g/kg MMDA for 35 days.

Point 3: Line 22: change birds

Response 3: Checked and done

Point 4: Line 26: According to the writing, the detoxifier carries the mycotoxins themselves ????

Response 4: The product tested was a multicomponent mycotoxin detoxifying agent (MMDA) containing modified zeolite (Clinoptilolite), Bacillus subtilis, B. licheniformis, Saccharomyces cerevisiae cell walls and silymarin. This products was tested as detoxifier of 0.5 mg/kg (0.5 ppm) ochratoxin A (OTA) and 1 mg/kg (1 ppm) T-2 toxin on broiler chickens diets.

The detoxifying product alone and included at 3g/kg  was tested in treatment T-2 (non-contaminated diet). The mycotoxins were added to treatments T-3 (without detoxifier), T-4 and T-5 (both with the detoxifier but at two levels of inclusion)

Point 5: Line 34: What is the contaminated control?

Response 5:  The contaminated control was Treatment 3 (0.5 mg/kg OTA and 1 mg/kg T-2 toxin added to the non-contaminated diet T-1)

Point 6: Line 115: But Soybean meal and wheat presented values for OTA and T2. Which???

Response 6:  The analytical values for OTA and T2 toxin determined in the  three main ingredients: wheat, maize and soybean meal were  below their quantification limits (<1.6 micrograms/kg for OTA and <9.6 micrograms/kg for T2 toxin).

Point 7: Line 145, 147, 262, 297, 341, 347, 348, 353: change bird by chicken

Response 7Checked and done

Point 8: Line 257-259: Both paragraphs indicate the same. UNIFY

Response 8: Checked and the second repetition was eliminated

Point 9: Line 261: Indicate in the material and methods that H / L is used to study stress

Response 9: in Material and Methods section it has been added: “The heterophil to lymphocyte ratio was evaluated as a measure of stress”.

Point 10: Line 282-285: Reflect the data in the table, or change the table type, because what is indicated is not understood.

Response 10: checked and changed including the BW values in the text: “ At 36 days of life, the final BW of birds fed contaminated diet plus 1 g/kg was 4.6% higher than the BW of birds fed control contaminated feed (1995 g vs.1907 g), and when the inclusion level of the MMDA Mycotoxin Binder was 3 g/kg the observed body weight increase was 7.3% (2046 g vs. 1907 g) (Table 7).”

Point 11: Line 286: its 17,5%

Response 11: Checked and done

Point 12: Line 296: delete in grams, because it’s not in the table

Response 12: Checked and done

Round 2

Reviewer 2 Report

In my opinion the revised paper merits the final acceptance.